# Quantitative Trait Locus Mapping for Fusarium Wilt Race 4 Resistance in a Recombinant Inbred Line Population of Pima Cotton (*Gossypium Barbadense*)

**DOI:** 10.3390/pathogens11101143

**Published:** 2022-10-03

**Authors:** Abdelraheem Abdelraheem, Yi Zhu, Jinfa Zhang

**Affiliations:** Department of Plant and Environmental Sciences, New Mexico State University, Las Cruces, NM 88003, USA

**Keywords:** cotton, fusarium wilt, resistance, quantitative trait locus

## Abstract

*Fusarium oxysporum* f. sp. *vasinfectum* (FOV) race 4 (FOV4) causes seedling death immediately after emergence, in addition to leaf chlorosis and necrosis, vascular discoloration, plant wilting, defoliation, and plant death at late stages. Breeding for FOV4 resistance is the most cost effective management method. In this study, 163 recombinant inbred lines (RILs) of FOV4-resistant Pima S-6 × susceptible 89590, together with the two parents (*Gossypium barbadense*), were artificially inoculated with FOV4 and assayed for resistance based on foliar disease severity ratings (DSR) at 30 days post inoculation (dpi) in two replicated tests in the greenhouse or controlled conditions. Significant genotypic variations were detected for FOV4 resistance in a combined analysis of variance. Although a significant genotype × test interaction was detected for DSR, the 10 most resistant RILs had significantly and consistently lower DSR than the susceptible parent in both tests. The heritability estimate for DSR was 0.65, indicating that two-thirds of the phenotypic variation for FOV4 resistance in this Pima RIL population was due to genetic factors. Based on 404 polymorphic SSR markers, five and four quantitative trait loci (QTL) on six chromosomes (c14, c17, c19, c21, c24, and c25) were detected in Tests 1 and 2, respectively, and each explained 15 to 29% of the phenotypic variation. Three QTL on c17, c24, and c25 were in common between the two tests, accounting for 60% and 75% of the QTL detected in Tests 1 and 2, respectively. The three QTL were also reported in previous studies and will be useful for marker-assisted selection for FOV4 resistance in Pima cotton.

## 1. Introduction

Extra-long staple cotton or Pima cotton (*Gossypium barbadense* L.) has superior fiber quality, including long, strong, and fine fibers and is grown in about a dozen countries including the U.S. [1]. Fusarium wilt (FW), caused by the soil-borne fungal pathogen *Fusarium oxysporum* f. sp. *vasinfectum* Atk. Sny & Hans (FOV), is one of the serious problems in cotton production in the U.S. and worldwide [2]. FOV decreases cotton yield by causing leaf wilting, chlorosis and necrosis, vascular discoloration, plant stunting, defoliation, and plant death [3,4]. In the U.S., FOV races 1, 2, 3, 4, and 8 were identified with race 4 (FOV4), reported first in California in 2000 [5] and most recently in Texas in 2018 [6] and New Mexico in 2019 [7,8]. Although Upland cotton is generally more resistant to FW than Pima cotton, Pima S-6 is resistant to FOV4 and has served as an important source of resistance to FOV4 in breeding and genetic studies [9,10,11,12]. Zhu et al. [13] recently showed that Pima S-6 and its derived resistant genotypes decrease the penetration of FOV4 into the root and prevent it from invading the vascular system. Intraspecific genetic and breeding populations using the FOV4-resistant Pima S-6 or its derived lines as a parent have been created to develop promising lines and cultivars with FOV4 resistance and to identify molecular markers associated with FOV4 resistance [9,10,11,12]. There have been numerous studies to investigate the genetic basis of FOV resistance in Upland cotton (*G. hirsutum* L.) using bi-parental and association mapping populations (see [3] for a review). A meta-analysis showed that most FOV resistance quantitative trait loci (QTL) were located on five chromosomes, i.e., c6, c14, c17, c22, and c25 [14]. Abdelraheem et al. [15] recently conducted a genome-wide association mapping in U.S. Upland cotton using high density genome-wide single-nucleotide polymorphic markers (SNPs) based on the CottonSNP63K array [16] and identified 13 FOV4 resistance QTL on six chromosomes (c8, c14, c16, c17, c18, and c19). However, only a few studies in mapping QTL for FOV4 resistance were conducted on Pima cotton. Using F_2_ populations, Ulloa et al. [10] reported a major gene/QTL for resistance to FOV4 on c14 in Pima S-6 based on a limited number of simple sequence repeat (SSR) markers. However, no RIL populations with Pima S-6 as a resistant parent were previously used to verify the results. Based on a RIL population of susceptible Pima S-7 × resistant Upland Acala NemX, two major QTL for FOV4 resistance on c14 and c17 were identified, also based on SSRs [17].

FW caused by FOV4 is an inoculum density and temperature-dependent fungal disease [18,19]. FOV4-caused plant wilt and death occurs under optimal and high inoculum density and low temperature (21–23 ℃) conditions. Zhang et al. [20] showed that disease incidence caused by FOV4 was similar under high and low temperature conditions, but the disease severity and plant mortality were significantly higher under the low temperature conditions. However, it is unknown if previous genetic studies in FOV4 resistance were performed under such conditions. In this study, we hypothesized that common and unique genes/QTL could be detected under both temperature regimes. To test this hypothesis, the objectives of this study were (1) to evaluate a Pima RIL population derived from a cross of FOV4-resistant Pima S-6 × FOV4-susceptible 89590 for FOV4 resistance using high and low temperature regimes under the greenhouse or controlled conditions and (2) to perform QTL mapping to identify genomic regions for FOV4 resistance in Pima S-6.

## 2. Materials and Methods

### 2.1. Development of the RIL Population

The RIL population was developed from a cross of FOV4-resistant Pima S-6 [21] with a susceptible Pima germplasm line 89590 [22], through a single-seed descent method [23]. F1 plants were self-pollinated to produce individual F2 plants which were subsequently advanced to RILs through several generations of repeated self-pollination on an individual F2 plant basis. Therefore, for most genes and DNA markers, an RIL was homozygous for one genotype or another (e.g., AA or aa; or A1A1 or A2A2), with a minimum heterozygosity (Aa or A1A2). Between the two parental lines, Pima S-6 is higher yielding, while 89590 has a sea-island parentage and is an extra-long staple germplasm with longer, stronger, and finer fibers [22].

### 2.2. Experimental Designs, Inoculation Methods, and Assessment of FOV 4 Resistance

Two replicated tests were conducted in the greenhouse or controlled conditions for FOV4 resistance in 2019. Seeds were planted in 4-inch plastic pots (in five hills with 2–3 seeds per hill for each line). A randomized complete block design with two replications for the RILs and the two parents was used in each test (10 plants genotype^−1^ replication^−1^). To understand if temperatures affected genotypic responses to FOV4 and the detection of QTL for FOV4 resistance, and to also compare results between potting soil and farm soil, two types of soil were used in this study based on Zhang et al. [20]. Test 1 was conducted using a commercial potting soil (Miracle-Gro Moisture Control Potting Mix 2 CF; Scotts Co., Marysville, OH, USA) under a high temperature (HT) regime (24–32 °C), while Test 2 was conducted in a FOV4 pre-infected farm soil and under a low temperature (LT) regime (20–21 °C). The greenhouse used natural sun light with no supplementary light provided. Daily irrigation and weekly application of fertilizer were used to manage the plants.

In both tests, however, artificial inoculations were made at the first true leaf stage with FOV4. A local FOV4 isolate, identified to be the most virulent in our previous study [8], was used in this study. Ten ml of 1 × 10^6^ spores mL^−1^ conidial suspension of FOV4 were inoculated to seedlings in each pot (10 plants pot^−1^) without root wounding, because FOV4 can penetrate the cotton root without the need of root-knot nematodes.

The disease severity ratings (DSR) were evaluated at 7, 14, 21, and 30 days post inoculation (dpi) using a 0–5 rating scale, similar to Zhang et al. [24] for evaluating resistance to Verticillium wilt, as the following: 

0 no symptom

1 one wilted cotyledon 

2 two wilted cotyledons or two cotyledons abscised

3 first true leaf wilted or three leaves including two cotyledons abscised

4 whole plant wilted or more than three leaves abscised

5 dead plant

### 2.3. Analysis of Variance

An average DSR was calculated on a plot (pot) basis for the subsequent analysis. The results for DSR from the two tests were subjected to a combined analysis of variance (ANOVA) using SAS v. 9.4 software (SAS Institute, Cary, NC, USA, 2012). Significance level was set at the *p* < 0.05 level for both the F (ANOVA) and *t* tests, i.e., the least significant difference (LSD) for mean separation. Broad-sense heritability (*h_b_^2^*) for DSR was estimated on a genotypic means based on ANOVA.

### 2.4. Linkage Mapping and QTL Analysis

The RILs were genotyped with a total of 403 SSR polymorphic loci which were produced from 200 SSR primer pairs, because each primer pair produced one to multiple polymorphic markers [23]. The primer sequences for the SSR markers are available from the CottonGen database (http://www.cottongen.org, accessed on 1 September 2022). DNA was extracted using a cTAB-based quick extraction method [25], followed by PCR and gel electrophoresis, as detailed by Abdelraheem et al. [23]. The linkage map was constructed using JoinMap version 4.0 software [26]. Inclusive composite interval mapping [27] was performed based on QTL IciMapping version 4.0 software (https://www.integratedbreeding.net/, accessed on 1 September 2022). The linkage map with QTL was visualized using MapChart [28]. 

## 3. Results and Discussion

### 3.1. Analysis of Variance of FOV4 Resistance Based on DSR

The RILs along with the two parents were evaluated in Tests 1 and 2 under a high temperature (HT, 24–32 °C) and a low temperature (LT, 21–22 °C) regime, respectively. A combined analysis of variance over the two tests detected a significant variation at the *p* < 0.01 level in genotype (G) for DSR at 30 dpi. The two tests (T) also significantly differed in DSR in that Test 2 at HT regime incurred a significantly higher DSR than Test 1 at LT (1.65 vs. 4.67). The G × T interaction was further detected for DSR, indicating that the overall performance of the RILs differed under HT and LT conditions (Table 1). The broad-sense heritability estimate for DSR was 0.65 based on the combined ANOVA from the two tests (Table 1), indicating that about two-thirds of the phenotypic variation was determined by unknown genetic factors to be determined through the subsequent QTL analysis in this RIL population. Therefore, genetic variation played a more important role than environment variation in DSR among the RILs in this population. 

### 3.2. Performance of the RILs for FOV4 Resistance

The FOV4 disease incidence (DI) ranged from 19.22 to 100.00% in Test 1 and from 59.82 to 100.00% in Test 2 (Table 2). The DI means were not statistically different between the two tests (82.2% for Test 1 vs. 93.8% for Test 2), although Test 2 had a higher DI. Under both HT and LT conditions, foliar symptoms due to FOV4 infections appeared in a few of seedlings exhibiting one or two wilting cotyledons within a week of artificial inoculation. However, very few seedlings died under the HT conditions with a mortality rate of 0.2%, as compared to 83.7% under LT at 30 dpi (Table 2). 

The Pima S-6 parent showed a consistent level of FOV4 resistance than the susceptible Pima line 89590 under both HT and LT conditions (Figure 1). Although the correlation coefficient for DSR between the two tests was not significant (r = 0.101, *p* > 0.05 at n = 161), the 10 most resistant lines had a DSR of 0.25–0.78 in Test 1 and 1.58–2.41 in Test 2, which were consistently and significantly lower than that for the 10 most susceptible lines (2.25–2.63 in Test 1 and 3.95–5.00 in Test 2) at *p* < 0.05 (Table 3). The most resistant RILs provide important improved lines for further studies and breeding efforts in Pima cotton.

### 3.3. QTL Analysis

The genetic linkage map for the Pima RIL population was reported previously by Abdelraheem et al. [23]. Briefly, a total of 31 linkage groups with 403 polymorphic SSR loci spanned a total genetic distance of 1,449 cM and were assigned to 26 chromosomes at an average genetic distance of 3.6 cM between two adjacent marker loci. The number of markers varied between chromosomes with a total of 201 and 193 loci in the A and D subgenomes, respectively.

A total of five and four major QTL (all from the D-subgenome), each of which explained 15.12 to 29.01% of the phenotypic variation (PVE), were detected in Tests 1 and 2, respectively (Figure 2). In Test 1, the five QTL were on five chromosomes (c14/D02, c17/D03, c19/D05, c24/D08, and c25/D13), and each carried 16.5 to 29.01% of PVE. In Test 2, the four QTL were on four chromosomes (c17/D03, c21/D11, c24/D08, and c25/D13), and each carried 15.12 to 21.50% of the PVE. In both tests, all the DSR-reducing resistance QTL alleles were from Pima S-6 (Table 4). Importantly, three QTL (on c17, c24, and c25) were in common between the two tests, accounting for 60 and 75% of the QTL detected in Tests 1 and 2, respectively (Figure 2). The major QTL identified on c17/D03 in both tests was anchored by two SSR markers (BNL3408a-200 and CIR347-280) within a 180 kb region (48.63–48.81 Mb). It carried 29.01% of the PVE for FOV4 resistance with a high LOD score (21.54). This QTL is consistent with a major gene/QTL previously identified in the FOV4- resistant Pima S-6 in another study using F_2_ populations [10]. A major resistance QTL for FOV4 and FOV7 resistance on c17 was also reported in resistant Upland cotton [15,17,29].

The second common QTL identified on c24/D08 in Test 1 overlapped with that in Test 2. Because both QTL in the two tests contained a common SSR marker BNL2961b-400 within a 200 kb region (56.77–56.97 Mb), they were considered a consistent QTL for FOV4 resistance. The QTL had a LOD score of 9.89–10.25 and explained 19.87% of the phenotypic variation in DSR. The third common QTL detected on c25/D13 was flanked by two SSR markers (MUSS122 and JESPR056) at the 52.85 Mb region in Tests 1 and 2, which carried 21.84% of the PVE with a LOD score of 13.54 in both tests (Table 3). 

Three unique QTL were detected in Test 1 or 2 only. The QTL on c14/D02 was detected in Test 1 only, and it was located in an extremely large region between 6.25 and 69.61 Mb. The QTL on c19/D05 was also detected in Test 1, and it was also mapped to a large region (10.63 Mb) of the chromosome between 10.56 and 21.19 Mb. The QTL on c21/D11 was detected in Test 2 only, and it was located within an 8.72 Mb region (17.52–26.24 Mb) of the chromosome. Because most of the QTL detected were within large chromosome regions containing numerous putative genes, no further analysis on candidate genes was performed in this study. To fine map consistent resistance QTL, more RILs and DNA markers should be used for candidate gene identification and validation.

In addition to a major FOV4 resistance QTL/gene (named *Fov4*) detected in the resistant Pima S-6, Mauricio et al. [10] reported several minor QTL on chromosomes c3, c6, c8, c14, c17, and c25 using three intraspecific (*G. hirsutum* × *G. hirsutum* and *G. barbadense* × *G. barbadense*) F_2_ populations, five interspecific (*G. hirsutum* × *G. barbadense*) F_2_ populations, and one RIL population derived from seven parents, including Pima S-6. This current study identified FOV4-resistance QTL on c14 and c19 in Test 1, on c21 in Test 2, and on c17, c24, and c25 in both tests. The major QTL/gene on c14 in Pima S-6 reported by Mauricio et al. [10] was not detected in Test 2 in this present study. However, both Maurcio et al. [14] and this study detected FOV4-resistance QTL on c17 and c25. To fine map these QTL and discern differences between QTL reported by different studies, large genetic populations of RILs should be developed for repeated evaluation for FOV4 resistance. Based on a meta-QTL analysis [3,14], chromosomes c6, c17, c18, c19, c24, and c25 had higher numbers of FOV (races 1, 4, and 7) resistance QTL in different Upland genetic backgrounds tested in different environments. Thus, the three common QTL (on c17, c24, and c25) detected in FOV4-resistant Pima S-6 in this study were consistent with these reported previously. They will be useful for marker-assisted selection for FOV4 resistance in cotton.

## Figures and Tables

**Figure 1 pathogens-11-01143-f001:**
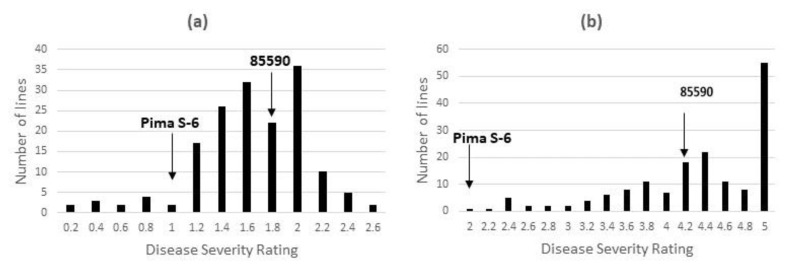
Disease severity rating (DSR) in the recombinant inbred line cotton (RIL) population of 163 lines from a cross of Pima S-7 × 89590 and the two parents at 30 days post inoculation in Test 1 under a high temperature regime (**a**) and Test 2 under a low temperature regime (**b**).

**Figure 2 pathogens-11-01143-f002:**
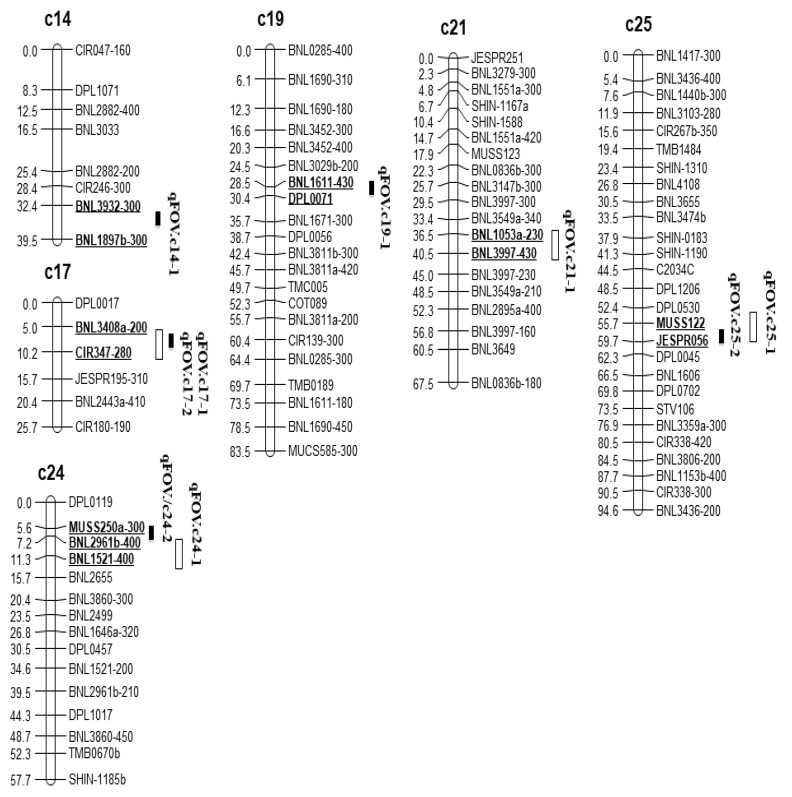
Quantitative trait loci (QTL) identified in the recombinant inbred line (RIL) cotton population of 163 lines from a cross of Pima S-7 × 89590 for Fusarium wilt race 4 (FOV4) resistance based on disease severity rating (DSR) at 30 days post inoculation.

**Table 1 pathogens-11-01143-t001:** A combined analysis of variance of Fusarium wilt race 4 (FOV4) resistance in the recombinant inbred line (RIL) cotton population of 163 lines from a cross of Pima S-7 × 89590 based on foliar disease severity rating (DSR) at 30 days post inoculation.

Sources of Variation	df	Mean Squares
Genotype	162	0.85 **
Test	1	0.21 **
Genotype × Test	162	0.52 **
Error	324	0.38
CV (%)		31.2
*h^2^*		0.65

** *p* < 0.01.

**Table 2 pathogens-11-01143-t002:** Mean disease severity rating (DSR), disease incidence (DI), and mortality due to Fusarium wilt race 4 (FOV4) infections in the recombinant inbred line (RIL) cotton population of 163 lines from a cross of Pima S-6 × 89590 at 30 days post inoculation in Tests 1 and 2.

Test	Mean DSR	Mean DI (%)	Mean Mortality (%)
1	1.65 a	82.18 a	0.20 a
2	4.67 b	93.81 a	83.65 b

Numbers followed by different letters indicate significant difference at *p* < 0.05.

**Table 3 pathogens-11-01143-t003:** Ten most resistant and 10 most susceptible lines to Fusarium wilt race 4 (FOV4) in the recombinant inbred line (RIL) cotton population of 163 lines from a cross of Pima S-7 × 89590 based on disease severity rating (DSR) at 30 days post inoculation in Test 1 and Test 2.

Line	Test 1	Test 2	Line	Test 1	Test 2
10 most susceptible lines			10 most susceptible lines		
NMPRIL21	0.25	1.58	NMPRIL6	2.63	3.95
NMPRIL20	0.38	2.06	NMRIL15	2.62	5.00
NMPRIL24	0.44	2.22	NMRIL140	2.50	4.67
NMPRIL135	0.46	2.34	NMRIL72	2.46	5.00
NMPRIL12	0.48	2.41	NMRIL67	2.41	5.00
NMPRIL7	0.50	2.05	NMRIL91	2.33	5.00
NMPRIL55	0.60	2.04	NMRIL82	2.29	4.86
NMPRIL65	0.67	2.01	NMRIL75	2.27	5.00
NMPRIL71	0.72	1.98	NMRIL101	2.25	4.83
NMPRIL100	0.78	2.01	NMRIL110	2.25	4.35
			LSD (0.05)	1.12	1.18

**Table 4 pathogens-11-01143-t004:** Quantitative trait loci (QTL) for Fusarium wilt race 4 (FOV4) resistance in the recombinant inbred line (RIL) cotton population of 163 lines from a cross of Pima S-7 × 89590 based on disease severity rating (DSR) at 30 days post inoculation.

QTL Name	Chr.	Left Marker	Right Marker	Physical Location (Mb)	LOD	PVE (%)	Direction
*qFOV.c14-1*	c14	BNL3932-300	BNL1897B-300	6.25–69.61	22.54	16.54	Pima S-6
*qFOV.c17-1*	c17	BNL3408a-200	CIR347-280	48.63–48.81	21.54	29.01	Pima S-6
*qFOV.c17-2*	c17	BNL3408a-200	CIR347-280	48.63–48.81	21.54	29.01	Pima S-6
*qFOV.c19-1*	c19	BNL1611-430	DPL0071	10.56–21.19	11.58	18.57	Pima S-6
*qFOV.c21-1*	c21	BNL1053a-230	BNL3997-430	17.52–26.24	9.98	15.00	Pima S-6
*qFOV.c24-1*	c24	MUSS260a-300	BNL2961b-400	56.77–56.97	10.25	19.87	Pima S-6
*qFOV.c24-2*	c24	BNL2961b-400	BNL1521-400	56.77–56.97	11.89	19.87	Pima S-6
*qFOV.c25-1*	c25	MUSS122	JESPR056	52.85	13.54	21.84	Pima S-6
*qFOV.c25-2*	c25	MUSS122	JESPR056	52.85	13.54	21.84	Pima S-6

Chr, chromosome number; PVE, phenotypic variation explained by the QTL.

## Data Availability

Not applicable.

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
