# Peer review of "Quantitative Trait Locus Mapping for Fusarium Wilt Race 4 Resistance in a Recombinant Inbred Line Population of Pima Cotton (Gossypium Barbadense)"

_pathogens, 2022, doi:10.3390/pathogens11101143_

Round 1

Reviewer 1 Report

File attached

Author Response

Dear Reviewer,

We thank you very much for your constructive review. The response to the comments is in the attached file.

Best wishes,

Jinfa Zhang, Ph.D.

Professor

New Mexico State University

Las Cruces, NM 88003

USA

Reviewer 2 Report

This study performed QTL mapping for FOV4 resistance in this Pima RIL population. Some issues should be addressed before further consideration.

  1. The environments for phenotyping were limited, it is better to added more. Moreover, two Test for phenotyping in this study with different temperature, why setting with two different temperatures? The soil of two Test with different treatments, one was with FOV4 pre-infected farm soil, the other was commercial potting soil, the phenotype of DSR in these two environments varied a lot, what’s the consideration about this? If set like this (one soil with FOV4 pre-infected and one with normal soil), the phenotype of resistance can be evaluated by difference of DSR between the two treatments?
  2. Several QTLs detected for FOV4, but without details in results and discussion, such as the physic region about the QTL region? The candidate genes? Need more deep disscussion
  3. Two “Table3 “in the MS, for the first Table3 DSR shows significant difference between Test1 and Test2, the mean for Test1 and Test2 was meaningless.
  4. For the second Table3, suggesting to add one column for environments (Tests), and the QTLs in same region use one common name.

Author Response

Dear Reviewer,

We thank you very much for your critical review. Revision has been made accordingly. Please see the attachment for our responses to your comments.

Best wishes,

Jinfa Zhang, Ph.D.

Professor

New Mexico State University

Las Cruces, NM 88003

USA

Round 2

Reviewer 1 Report

Looks fine.

Author Response

We thank the reviewer very much for the support and comment. The final version of MS is now polished for readability.

Reviewer 2 Report

The comments have been addressed significantly. I suggested to accept after minor revision for some text editing and English improving.

Author Response

We thank the reviewer very much for the comment and support! The final version is now polished with a clean copy submitted.